# Transformer-Based Approach to Pathology Diagnosis Using Audio Spectrogram

**Mohammad Tami** [1], **Sari Masri** [1], **Ahmad Hasasneh** [1] and **Chakib Tadj** [2,*]

1   Department of Natural, Engineering and Technology Sciences, Faculty of Graduate Studies, Arab American University, Ramallah P.O. Box 240, Palestine; m.abutami@student.aaup.edu (M.T.); s.masri3@student.aaup.edu (S.M.); ahmad.hasasneh@aaup.edu (A.H.)
2   Department of Electrical Engineering, École de Technologie Supérieur, Université du Québec, Montréal, QC H3C 1K3, Canada
*   Correspondence: chakib.tadj@etsmtl.ca

**Abstract:** Early detection of infant pathologies by non-invasive means is a critical aspect of pediatric healthcare. Audio analysis of infant crying has emerged as a promising method to identify various health conditions without direct medical intervention. In this study, we present a cutting-edge machine learning model that employs audio spectrograms and transformer-based algorithms to classify infant crying into distinct pathological categories. Our innovative model bypasses the extensive preprocessing typically associated with audio data by exploiting the self-attention mechanisms of the transformer, thereby preserving the integrity of the audio's diagnostic features. When benchmarked against established machine learning and deep learning models, our approach demonstrated a remarkable 98.69% accuracy, 98.73% precision, 98.71% recall, and an F1 score of 98.71%, surpassing the performance of both traditional machine learning and convolutional neural network models. This research not only provides a novel diagnostic tool that is scalable and efficient but also opens avenues for improving pediatric care through early and accurate detection of pathologies.

**Keywords:** pathology diagnostics; infant cry classification; audio spectrograms; transformer; pediatric healthcare





## 1. Introduction

Hundreds of millions of infants are born every year, and millions of them develop diseases shortly after birth [1]. This results in an increase in the number of deaths each year. Detecting and recognizing these diseases can be challenging because infants cannot communicate or react like adults, making it difficult to provide the necessary care and treatment. It has been shown that sepsis and respiratory distress syndrome (RDS) are the most common pathologies associated with infant deaths worldwide [2–4]. Thus, this study aims to diagnose these pathologies in their early stages by analyzing crying signals and predicting their onset.

In particular, sepsis is one of the most common diseases affecting infants, with 48.9 million cases reported annually. Each year, 9 million newborns are affected by sepsis, a blood infection that occurs in infants younger than 12 weeks. Sadly, 11 million infants die each year from this disease, accounting for nearly 20% of all deaths worldwide [3]. Sepsis can be difficult to diagnose because its symptoms can mimic those of other diseases. Blood tests to check white blood cell count and C-Reactive Protein (CRP) levels are the first step in diagnosing this infection. Blood cultures should also be performed to detect the bacteria or fungus causing the infection. This process can be challenging due to time constraints and small sample sizes. In addition, urine tests should be conducted to check for urinary tract infections. Finally, clinical evaluation of the mother during pregnancy is also necessary [5].

In addition, respiratory distress syndrome (RDS) is a common condition affecting infants born before 37 weeks. It is estimated that approximately one million babies worldwide

die due to RDS pathology. RDS typically occurs in infants born before their due date, usually before 28 weeks [4]. It often occurs when the baby's lungs are not fully developed and cannot provide enough oxygen to the body. RDS can cause breathing failure in newborns. Although babies with RDS can survive, they require special medical care. Diagnosing this pathology requires complex procedures, including X-rays, computed tomography (CT) scans, electrocardiograms, and frequent blood tests to monitor the oxygen levels in the blood [6].

Therefore, the diagnosis of RDS and sepsis requires additional healthcare and is not an easy task. Early detection of these and other hidden illnesses could reduce the number of deaths and provide better care for infants in their early stages [7]. In addition, it could be more effective if infants could communicate with their environment to better understand their situation. Therefore, parents and experts are developing ways to better understand the needs of infants by observing their cries. However, understanding cries can be challenging for untrained parents and clinicians. To reduce false positive results from manual diagnosis, it is important to develop better methods for understanding infant communication [8]. The researchers focused on conducting a deep analysis of the infant cry signal to better understand infant needs and illnesses. This could potentially help avoid complicated tests and be conducted without the need for specialists.

Researchers have recently discovered that it is possible to analyze cries and associate them with diseases [9], and spectrogram waves have been found to be different for infants with different diseases [10]. Analyzing these waves without the use of artificial intelligence (AI) could be challenging. However, by employing machine learning (ML) and deep learning (DL) models, it is possible to accurately classify different diseases. This topic has received considerable attention due to the significance of illness classification. Infant crying signals have distinct spectrogram shapes that can be associated with various common diseases, such as sepsis and respiratory distress syndrome (RDS). Therefore, this research focuses on analyzing spectrogram images to achieve high accuracy.

The contributions of this research study include diagnosing sepsis and RDS by analyzing infant cry signals using a pre-trained spectrogram transformation model without preprocessing the spectrogram images of the waves, as this is part of the modeling process. Second, this study utilizes only the spectrogram without extracting complicated features such as Gammatone Frequency Cepstral Coefficients (GFCCs) and heart rate (HR) from the voice or building a complex model. This approach aims to achieve a more curated result and reduce false positives and true negatives, ultimately providing support to medical clinics and hospitals as desired specifically in a low-resource environment.

The remainder of this paper is structured to provide a comprehensive exploration of the topic at hand, beginning with a detailed literature review that contextualizes our research within the existing body of knowledge. Following this, Section 3 illustrates the dataset selection process and the experimental procedures employed, ensuring the transparency and reproducibility of our research methodology. Subsequently, the findings are presented in Section 4, providing a thorough analysis and interpretation of the data collected. This paper concludes with Section 6, in which we summarize the main findings, reflect on the implications of our results, and propose directions for future research directions, with the aim of paving the way for further scientific investigation on this research topic.

## 2. Literature Review

The phenomenon of an infant crying has captured the interest of many researchers [11–14], resulting in a variety of studies exploring different approaches. Some work has focused on identifying specific pathologies such as sepsis [8,15], respiratory distress syndrome (RDS) [8,15], asphyxia, autism spectrum disorder (ASD) [12], and hypo-acoustic [13]. Other studies have performed binary classification to compare the health of infants with specific pathologies [14]. These research studies consisted of two main phases: feature computation and feature extraction. They used the CAS (Context-Aware System) and focused on

different audio domains such as cepstral domain features, prosodic domain features, image domain features, time domain features, and wavelet domain features [16]. Then, some numerical features were used in machine learning models and other complementary features such as spectrogram were processed using deep learning models, and some researchers used combined features and models such as convolutional neural network (CNN) with linear regression [8,12,15].

Many researches focus on extracting cepstral domain features such as Mel Frequency Cepstral Coefficients (MFCCs) [15,17,18], Linear Frequency Cepstral Coefficients (LFCCs) Short Time Cepstral Coefficients (STCCs) [19], and Bark Frequency Cepstral Coefficients (BFCCs) [20] in audio signal feature extraction and feeding them into machine learning (ML) and deep learning models; for example, in [21], they used MFCC in classifying infant crying causes with 76% accuracy by using ML learning algorithms such as K-Nearest Neighbor (KNN), Support Vector Machine (SVM), and Naive Bayes Classifier (NBC); however, in the study [17], they built binary classification for healthy and pathological infants, achieving 91% by combining the MFCC feature extracted with the Hidden Markov Model (HMM).

In addition, other studies have utilized prosodic domain features, which encompass variations in intensity, fundamental frequency, and other aspects that may be significant in audio signal analysis. For instance, in [22], researchers used the mean, median, standard deviation, minimum, and maximum values of F0 and F123. Other studies have combined cepstral features with prosodic domain features, as in [23], where prosodic and MFCC features were used in conjunction with a deep learning model, resulting in a remarkable accuracy rate of 96%. This suggests that the use of multiple domain features can improve the results.

The use of the discrete wavelet transform (DWT) is another method for feature extraction. It helps to extract coefficient features. In a study [24], this approach was investigated for its effectiveness in classifying different types of crying to predict the needs of the infant, such as hunger, sleep, discomfort, and others. A single-layer neural feed-forward (SLNFF) was used and an average accuracy of 80% was reported for different types of classifications.

Another important feature of sound signals is their spectrogram images, which can be full of character and provide valuable information. An image or time-frequency representation of audio can be generated using techniques such as the short-time Fourier transform (STFT) [16]. The authors in [23] proposed to use the spectrogram along with CNN models to classify the situation of the infant as either sleepy or in pain, achieving an accuracy of over 78%. Also, the authors in [25] used the spectrogram to classify the cry signal into different categories, such as pain, hunger, and sleepiness, and they achieved 88.89% accuracy by using the spectrogram and feeding it into a CNN model with the radial basis function. Another study was proposed in [26] and achieved 71.68% accuracy by using the spectrogram with SVM.

On the other hand, it can be seen that some researchers have focused on combining different modalities of features together; for example, in [8], the authors proposed to combine GFCC and HR features on sepsis and RDS using ML and SVM and achieved 95.29% accuracy, while the authors in [15] proposed a combined deep learning model of GFCC, HR, and spectrogram features and achieved 97. 5% accuracy. Also, the authors in [27] proposed to combine spectrogram with prosodic feature with a CNN model and achieved 97% accuracy. Moreover, the authors in [28] combined MFCC, chromogram, Mel-scaled spectrogram, spectral contrast, and Tonnetz features and used the DNN model, achieving 100% accuracy in classifying normal and suffocating cries and 99.96% accuracy for classifying non-suffocating and suffocating cries.

Although numerous researchers in the field of audio processing for medical diagnostics have reported impressive classification performance with their models, there is still a need to investigate the complex processing that must be applied to the raw audio to achieve such high accuracy. In this proposed study, the authors employ a straightforward audio representation using spectrograms, coupled with a transformer-based model that processes these spectrograms as image patches. This approach is designed to capture the internal

complexity of the audio representation through a spectrogram, which helps the model achieve high classification results. To the best of the authors' knowledge, this method of processing audio data for medical diagnostics is pioneering in its use of CNN-free models. The proposed approach relies solely on the use of a spectrogram from the infant cry audio dataset instead of using multiple features such as GFCC and HR to achieve high performance results.

The main contributions of this study are threefold. First, it introduces a simple, yet effective, model for audio processing in medical diagnostics, emphasizing the importance of simplicity in model design. Second, it innovates by utilizing spectrograms in conjunction with a transformer-based model, processing these spectrograms as image patches, which is a novel approach in the field. This technique leverages the transformer's ability to handle complex data structures, thereby capturing the intricate patterns within audio signals more efficiently. Finally, this study demonstrates improved accuracy in classification performance, setting a new benchmark for CNN-free models in medical diagnostics by effectively processing audio data without the need for extensive preprocessing or feature extraction. This advancement underscores the potential of transformers to revolutionize audio-based medical diagnostic processes.

## 3. Materials and Methods

In this research, a comprehension methodology was implemented to ensure reliable and robust results. Figure 1 outlines the four main stages taken to conduct this research. First, audio data of the crying infants was gathered and then segmented. This segmented raw audio served as the basis for the construction of spectrograms. These spectrograms were then fed into a transformer, where a multiclass classifier was trained. Finally, the model was evaluated using various metrics to formulate a conclusion regarding the model's performance and its efficiency.

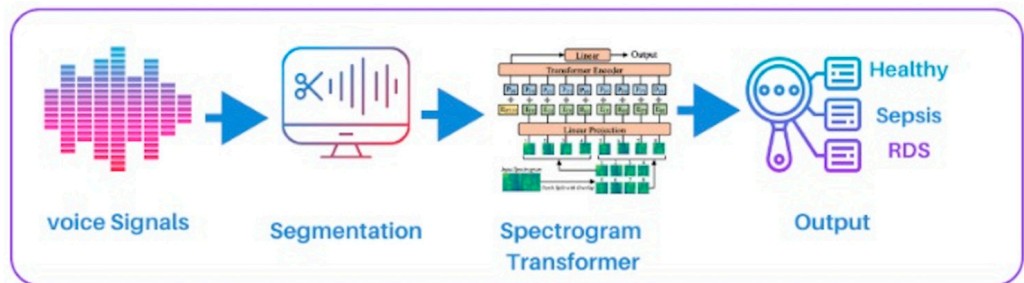

**Figure 1.** The workflow of the proposed model for classifying infant pathological cries.

### 3.1. Dataset Description

The dataset for this study was obtained from the Saint-Justine Children's Hospital in Montreal, Canada, and the Al-Raee and Al-Sahel Hospitals in Lebanon, which has recently been used in several related studies [8,15,29]. It consists of audio samples of neonatal crying from infants ranging in age from 1 to 53 days, with diverse demographic backgrounds, as detailed in Table 1. There was a total of 53 recordings from 17 newborns for the sepsis group, 102 recordings from 33 newborns for the RDS pathology group, and 181 recordings from 83 newborns for the healthy group. Recordings were captured using a standard digital 2-channel Olympus recorder with 16-bit resolution and a sampling rate of 44,100 Hz, positioned 10 to 30 cm from the infants. The infants' health conditions were determined by medical examinations, and the cries were categorized accordingly as indicating a specific pathology or normal. The dataset includes various pathologies, including respiratory distress syndrome (RDS), kidney failure, aspiration, asphyxia, and sepsis, and non-pathological cries. Each original audio recording lasted an average of 90 s and was recorded five times for each infant.

**Table 1.** Description of datasets for selected pathological cases.

| Demographic Factors | Details |
| --- | --- |
| Gender | Female and male |
| Babies' ages | 1 to 53 days old |
| Weight | 0.98 to 5.2 kg |
| Origin | Canada, Haiti, Portugal, Syria, Lebanon, Algeria, Palestine, Bangladesh, and Turkey |
| Race | Caucasian, Arabic, Asian, Latino, African, Native Hawaiian, and Quebec |

The challenge of a limited number of recordings, due to factors such as the unpredictability of capturing certain pathologies during data collection, the extensive process for ethical and technical approval, potential loss of samples, and the need for guardian consent for inclusion in the database, was addressed. To mitigate data scarcity, each recording was segmented into multiple expiration segments to enrich the analysis of pathological infant crying. This segmentation resulted in the creation of multiple expiration (EXP) segments. The dataset was then evenly sampled to ensure an equal representation of samples from each category. This approach yielded a balanced and uniform dataset containing 3900 segmented records with 1300 samples in each class, as shown in Figure 2.

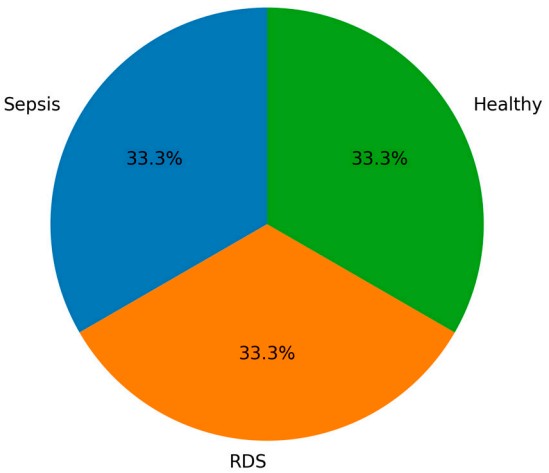

**Figure 2.** Distribution of the three pathologies' samples. It shows that the three diseases are equally represented.

### 3.2. Data Preprocessing

The infant cry dataset was preprocessed by previous researchers [8,15,29] to eliminate any periods of silence and to filter and segment each recording. The recordings were segmented, and various labels were applied; for example, expiratory cries were labelled as EXP, while voiced inspiratory cry segments were labelled as INSV. This labeling was facilitated by the use of WaveSurfer software (1.8.8). In our research, we specifically used the EXP segments from each cry recording, treating each one as an individual sample. As mentioned earlier, a major challenge in biomedical research is the scarcity of data, where the incidence of certain pathologies in newborns is unpredictable. By segmenting each cry signal, we have been able to partially overcome this problem. Moreover, we excluded all segments shorter than 200 milliseconds, as they did not provide sufficient information to analyze the infant's cry signals.

*3.3. Data Transformation*

To adapt the raw audio for effective processing by the Audio Spectrogram Transformer model [30], three key transformations were applied. The initial transformation involved resampling the audio to a 16 kHz frequency. This step is crucial for striking a balance between retaining audio quality and optimizing computational efficiency, a principle supported by references [30–32]. Resampling at this frequency ensures the audio data maintain enough detail for the model's requirements while minimizing the computational resources needed for processing.

The second transformation applied was padding the audio clips to ensure uniform lengths. Unlike truncation, which removes parts of the audio, padding involves adding silence to the audio clips that do not meet a certain length requirement. This process is essential to create a consistent input size across all audio samples, facilitating the model's ability to learn from and generalize across the dataset effectively.

Finally, the third transformation converted the raw audio signals into spectrograms, utilizing specific parameters to ensure a detailed and consistent analysis of the audio. Spectrograms offer a two-dimensional representation of sound, illustrating the spectrum of frequencies as they change over time. This conversion employs an FFT (Fast Fourier Transform) length of 2048, a Hann window type, a window length equal to the FFT length (2048), and a hop length (window shift) of 512. Each of these parameters plays a vital role in constructing an effective spectrogram for infant cry audio classification with a transformer model. The longer FFT length provides a more detailed frequency resolution, capturing subtle differences in frequency components crucial for identifying patterns within infant cries. The Hann window minimizes spectral leakage, enhancing the clarity of the time–frequency representation, which is essential for the model to learn effectively. Matching the window length to the FFT length ensures comprehensive frequency analysis, allowing for the full representation of audio segments. A shorter hop length offers higher time resolution, giving detailed temporal information that aids in understanding how the frequency content changes over time, a critical factor in distinguishing different cries.

This methodological approach is advantageous for several reasons. First, it encapsulates both temporal and frequency information in a visually intuitive format, facilitating easier pattern identification and analysis within the audio data for the model. Secondly, by emphasizing the nuances in the audio that might not be apparent in the raw waveform, spectrograms enhance the model's accuracy in classification or analysis tasks. By converting audio into spectrograms after ensuring all clips are of uniform length through padding and resampling for optimal quality and efficiency and by finely tuning these specific parameters, the process significantly boosts the model's performance. This provides a rich, detailed, and uniform representation of the audio, enabling the transformer model to leverage high-quality data for more nuanced distinctions between different classes of audio signals.

*3.4. Audio Spectrogram Transformer Architecture*

The fundamental idea behind the Audio Spectrogram Transformer (AST) for infant cry analysis is to adapt the principles of the Vision Transformer (ViT) for use with audio data [30,33]. In the ViT model, images are processed as patches, which are individually linearly embedded before positional embeddings are incorporated. These steps facilitate the retention of spatial relationships within the image. The assembled sequence of vectors is then fed into a transformer encoder, which employs self-attention mechanisms to model complex interactions between different parts of the image. The output of the encoder is a higher-dimensional representation of the original image, capturing both local and global dependencies.

Transformers, a class of deep learning models, offer several advantages over classical machine learning techniques. These include their ability to handle sequential data without the need for pre-defined temporal feature extraction, making them inherently adept at processing time-series data, natural language, and, as adapted in the AST, audio signals.

Additionally, transformers excel in parallel processing, allowing for significantly faster training times compared to recurrent neural networks (RNNs) and Long Short-Term Memory (LSTM) networks. Their self-attention mechanism provides an efficient way to capture dependencies regardless of their distance in the input sequence, leading to improved performance on tasks requiring an understanding of long-range interactions.

Transformers have found applications across a wide range of fields, showcasing their versatility and effectiveness. In natural language processing (NLP), models like BERT (Bidirectional Encoder Representations from Transformers) [34] and GPT (Generative Pre-trained Transformer) [35] have set new standards for tasks such as text classification, machine translation, and text generation. In computer vision, the Vision Transformer (ViT) model has demonstrated that transformers can achieve state-of-the-art results on image classification tasks without relying on conventional convolutional neural networks (CNNs) [36]. In the realm of audio processing, transformers have been applied to speech recognition and music generation [33,37,38], illustrating their capability to model complex audio patterns effectively. More recently, the adaptation of transformers for analyzing audio data through the Audio Spectrogram Transformer (AST) indicates the model's potential for biomedical applications and signal analysis [39], opening new avenues for research and practical applications in healthcare and beyond.

The primary aim of this study is to use the pre-trained Audio Spectrogram Transformer (AST) model and fine-tune it with health audio data for solving a multiclass classification task. The decision to employ Audio Spectrogram Transformer (AST) models was based on their proven effectiveness at processing complex audio patterns. This ability is particularly beneficial when dealing with the intricate and subtle variations found in health-related audio data. Recent research leveraging AST models with spectrograms has shown promising results, as evidenced in recent studies [40–42]. This success has led us to apply the AST model to health-related audio data, which is, to the best of our knowledge, its first application in the health field. The authors aim to leverage its superior pattern recognition capability for a multiclass classification challenge by fine-tuning the AST model with specific health-related audio data. This strategy not only exploits the unique strengths of AST for processing audio data, but also explores its potential to significantly improve the analysis and categorization of audio datasets in the health domain.

The AST model processes spectrograms through self-attention mechanisms, allowing it to focus on relevant parts of the input without losing information, unlike CNNs which can lose fine details due to spatial reduction through convolution and pooling layers. Transformers compute relationships between all parts of the input simultaneously, efficiently capturing both global and local dependencies. This ability to maintain a comprehensive understanding of the entire input at each layer enables more nuanced and detailed modelling of audio data, leading to its superior performance.

The original AST model consists of a process where an audio waveform is transformed into a $128 \times 100$ t spectrogram, which is then divided into $16 \times 16$ patches. These patches are flattened into 1D embeddings and combined with positional embeddings. The sequence, appended with a [CLS] token, is processed by a standard transformer encoder with 12 layers and 12 heads, designed for classification. The output of the [CLS] token, transformed by a linear layer with sigmoid activation, serves as the basis for classification. This approach leverages transformer's capabilities for audio classification, emphasizing ease of implementation and suitability for transfer learning. In the following subsection, we will describe the internal structure of the Audio Spectrogram Transformer (AST), as shown in Figure 3.

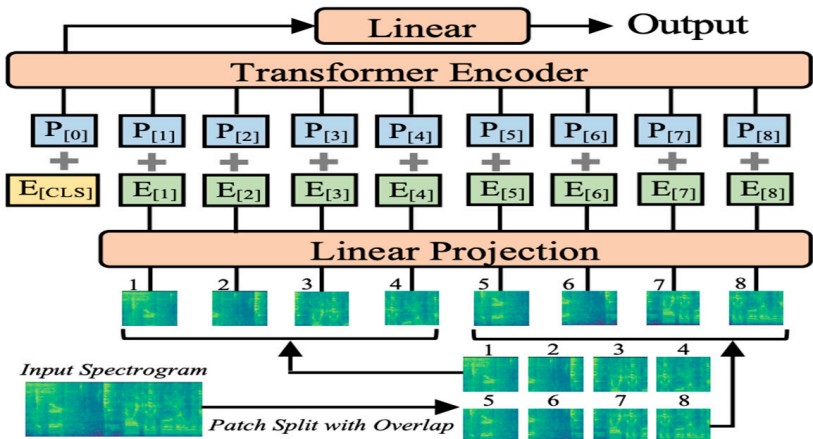

**Figure 3.** The proposed Audio Spectrogram Transformer (AST) architecture. The 2D audio spectrogram is split into a sequence of 16×16 patches with overlap and then linearly projected to a sequence of 1D patch embeddings. Each patch embedding is added with a learnable positional embedding. An additional classification token is prepended to the sequence. The output embedding is input to a transformer, and the output of the classification token is used for classification with a linear layer [36].

### 3.4.1. AST Spectrogram Generation

In the initial phase, the cry of the infant is recorded and transformed into a spectrogram within the Audio Spectrogram Transformer (AST) model, as illustrated in Figure 4. This transformation process converts the raw audio signals into a visual format that maps sound frequency, intensity, and duration onto a two-dimensional plane, effectively creating an image-like representation of the sound. Spectrogram conversion is a critical preprocessing step that enables the application of image-based processing techniques to audio data. This approach is advantageous for several reasons. First, spectrograms provide a more intuitive visualization of the sound's characteristics, such as its frequency content over time, which can be crucial for detecting patterns or anomalies. Second, by representing audio signals in this visual form, it becomes possible to leverage advanced image processing algorithms and machine learning models that are typically designed for image data, thus opening a wider range of analytical techniques. Essentially, the use of spectrograms facilitates a more effective analysis by bridging the gap between the complex, time-varying nature of audio signals and the powerful, well-established domain of image analysis.

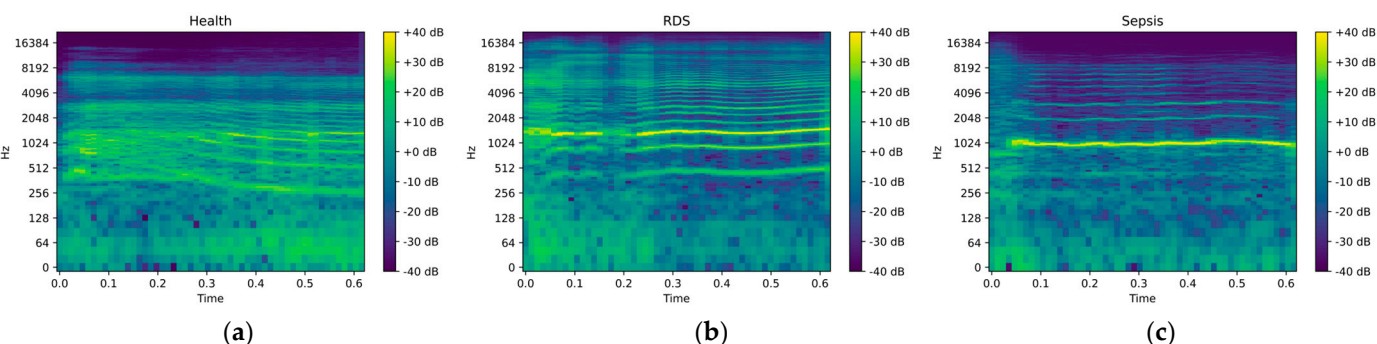

**Figure 4.** Spectrogram of the audio data where (**a**) is healthy class, (**b**) is RDS class, and (**c**) is sepsis class respectively.

### 3.4.2. Patch-Based Input Representation

After constructing the spectrogram from the audio, the spectrogram is segmented into a series of patches. This method, inspired by how vision transformers process image data, allows the model to focus on localized frequency–time regions within the spectrogram.

Each patch encapsulates specific features of the cry, such as pitch variations or intensity fluctuations, which are critical for diagnostic purposes.

### 3.4.3. Transformer Encoder

The transformer-based encoder model, at the heart of this architecture, processes these patches sequentially. Unlike traditional convolutional neural networks, which process images as a whole, the transformer model treats each patch as a part of a sequence. This sequential processing enables the model to understand the temporal dynamics within the cry, which is pivotal for identifying subtle acoustic markers that indicate various health conditions.

### 3.4.4. Linear Layer

The final output of the model produces a set of probabilities using sigmoid activation linked to different infant conditions, derived from the cry analysis. This output serves as an input layer for a subsequent classification task, representing an additional feature space of the input data utilized in the classification process. The transformer's capability to identify complex patterns in the spectrogram achieves a diagnostic accuracy beyond what conventional audio processing methods could attain, enhancing the precision of the classification.

### 3.5. Model Training and Evaluation Measures

Training was performed using a Colab notebook equipped with an A100 GPU, and the time required to fine-tune the model on the selected dataset was approximately 15 min. The AST model was developed using the transformers library, a comprehensive package of various cutting-edge pre-trained models. The audio dataset was segmented into training, validation, and testing parts, with distributions of 70%, 15%, and 15%, respectively.

The selection of model hyperparameters is crucial for achieving high-quality training results. In this context, Categorical (Multiclass) Cross-Entropy was chosen as the utilized loss function due to its effectiveness in handling multiclass classification problems, which are inherent in our audio dataset. This loss function is particularly suitable for scenarios where each input is intended to be classified into one out of many possible categories. It works by comparing the predicted probability distribution across all classes with the actual distribution, where the true class has a probability of 1 and all others have a probability of 0. This approach incentivizes the model to not only predict the correct class, but to also be confident in its prediction, thereby enhancing the overall accuracy of the model.

$$Categorical\ Cross-Entropy\ Loss = -\sum_{c=1}^{M} y_{o,c} \log(p_{o,c}) \tag{1}$$

where $M$ is the number of classes, $y_{o,c}$ is the ground truth (actual value, 0 or 1, indicating whether $c$ is the correct class for observation $o$), $p_{o,c}$ is the predicted probability of observation $o$ being of class $c$. The negative sign is used to make the overall loss positive, as the log function yields negative values for inputs between 0 and 1.

Furthermore, numerous tests were carried out to identify the optimal hyperparameters. A learning rate of $6 \times 10^{-5}$ was chosen, probably due to its balance between speed and accuracy in the gradient descent, preventing overshooting of the global minimum. The use of a linear scheduler to gradually adjust this learning rate likely helped to fine-tune the training process as the model approached convergence, improving the precision of the updates. The rapid convergence of the model convergence within 6 epochs, as shown in Figure 5, suggests an efficient learning process. Furthermore, the implementation of a 0.5% weight decay was particularly significant. This weight decay, which acts as a form of L2 regularization, not only helps to prevent overfitting and ensures that the model generalizes well to new data, but also stabilizes the training process by preventing the weights from growing excessively large. In addition, choosing a batch size of 16 helps maintain a balance between computational efficiency and the ability to capture sufficient data variance in

each training step. Empirically, these hyperparameter settings appeared to be optimal for our audio dataset, allowing fast learning while maintaining the quality of the model's predictions (Table 2).

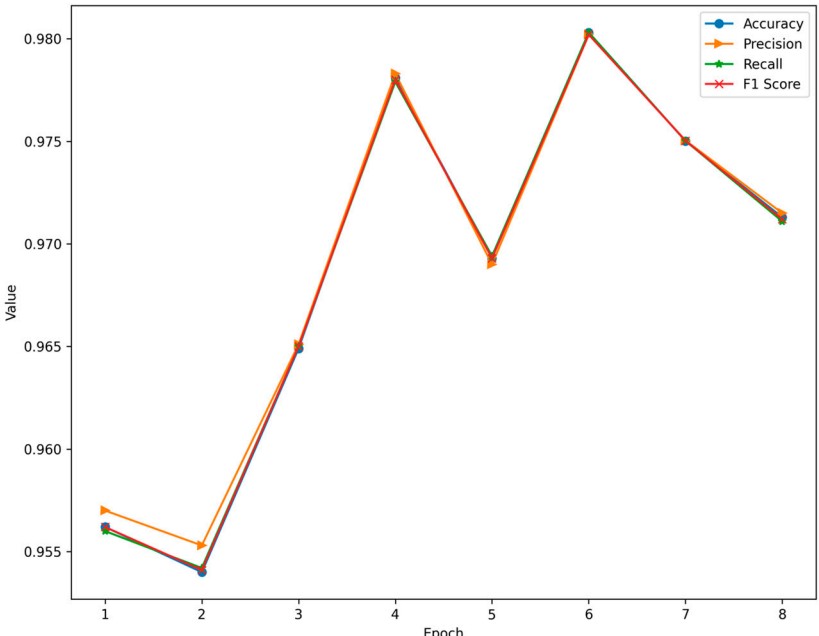

**Figure 5.** Progression of the best model validation metrics over epoch.

**Table 2.** Hyperparameters of the transformer model.

| Hyperparameter | Best Value |
|---|---|
| Epoch | 6 |
| Learning rate | $6 \times 10^{-5}$ |
| Learning rate scheduler | Linear |
| Weight decay | 0.5% |
| Batch size | 16 |
| Loss function | Categorical Cross-Entropy |
| Optimizer | adamW |

To further assess the model performance, four key classification metrics were computed: accuracy, recall, precision, and F1 score, as described in Equations [2–5]. Accuracy, defined as the proportion of true results, both true positives (TP) and true negatives (TN), among the total number of cases examined, provides a general measure of the model performance. Recall, also known as sensitivity, measures the ability of the model to correctly identify positive results from all true positive cases. Precision, on the other hand, reflects the model's ability to correctly predict positive outcomes out of all predicted positives. Finally, the F1 score, which is the harmonic mean of precision and recall, provides a balance between the two, providing a comprehensive measure of the model's accuracy, especially in situations where the class distribution is unbalanced. These metrics collectively provide a thorough assessment of the model's classification capabilities.

$$Accuracy = \frac{TP + TN}{TP + TN + FP + FN} \tag{2}$$

$$Recall = \frac{TP}{TP + FN} \tag{3}$$

$$Precision = \frac{TP}{TP + FP} \tag{4}$$

$$F_1 = 2 \times \frac{\text{Precision} \times Recall}{\text{Precision} + Recall} \tag{5}$$

where *TP* represents true positives, *TN* true negatives, *FP* false positives, and *FN* false negatives.

To guide the hyperparameter tuning process, we observed the accuracy, recall, precision, and F1 score over the validation split during training. Figure 5 shows the increase in these metrics until they reach the peak in epoch 6. All metrics showed an ascending trend over the epochs, ultimately reaching their peak by the sixth epoch. This consistent improvement indicates that the model's ability to generalize improved with each epoch, until it stabilized at its best performance by the end of the observed training period. To avoid overfitting, an early stopping strategy was applied where the training stopped if the metrics of validation split decreased for two consecutive epochs.

Hyperparameter sensitivity analysis of audio spectrogram models reveals the nuanced effects of learning rate, epoch duration, learning rate scheduler, and weight decay on model performance, particularly in terms of the misclassification rates observed in the confusion matrices as illustrated in Figure 6.

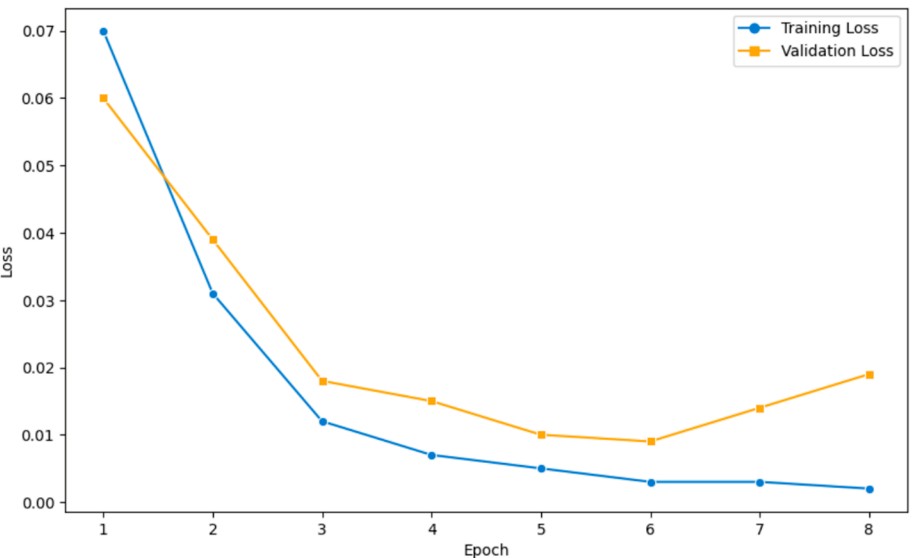

**Figure 6.** Training and validation loss per epoch.

For the models tested, varying hyperparameters led to distinct accuracy levels and misclassification patterns across different conditions (healthy, RDS, and sepsis) (Table 3). The first model, with a learning rate of $6 \times 10^{-5}$, epochs set to 6, a linear learning rate scheduler, and a weight decay of 0.5%, achieved the highest accuracy (98.69%). This configuration resulted in the fewest misclassifications, suggesting that a lower learning rate combined with a moderate number of epochs and a linear scheduler is highly effective for this task. The gradual reduction of the learning rate likely allowed for more precise adjustments in the model's weights, leading to better generalization on the testing data. The second model, with a slightly higher learning rate of $8 \times 10^{-5}$, fewer epochs (5), and similar scheduler and decay settings, also performed well (95%), but with a slight increase in misclassification rates. This indicates that while a higher learning rate can still yield high accuracy, it may lead to slightly more errors, possibly due to less stable convergence in the optimization landscape. The third model, with a significantly lower learning rate of $1 \times 10^{-5}$ and the same epoch count and other settings, showed a decrease in performance (93%). This suggests that too low a learning rate might not be sufficient to escape local minima or to converge to the global minimum within a limited number of epochs, leading to higher misclassification rates. The fourth model, utilizing a much higher learning rate of $5 \times 10^{-4}$, a constant learning rate scheduler, and the same weight decay, had the lowest

accuracy (87%). The constant learning rate likely contributed to this drop in performance, as it does not allow for the fine-tuning of model weights in the later stages of training, resulting in a higher number of misclassifications. This model demonstrates the risks of using a high learning rate with a constant scheduler, which can prevent the model from adjusting to more nuanced data features, leading to poorer generalization.

**Table 3.** Comparison of model performance based on hyperparameter configurations.

| Model | Learning Rate | Epochs | Scheduler | Weight Decay | Accuracy |
|---|---|---|---|---|---|
| 1 | $6 \times 10^{-5}$ | 6 | Linear | 0.5% | 98.69% |
| 2 | $8 \times 10^{-5}$ | 5 | Linear | 0.5% | 95% |
| 3 | $1 \times 10^{-5}$ | 5 | Linear | 0.5% | 93% |
| 4 | $5 \times 10^{-4}$ | 5 | Constant | 0.5% | 87% |

## 4. Experimental Results

The rapid convergence of the proposed model within six epochs, as shown in Figure 6, suggests an efficient learning process. The graph illustrates the training and validation loss of an Audio Spectrogram Transformer model over six epochs. The rapid convergence of the model is evident, taking only six epochs to reach a state of minimal loss. This is a strong indication of the model's ability to learn and generalize from the data efficiently.

Observing the trends in the graph, both training and validation losses decrease sharply and steadily as the number of epochs increases, which is a good sign of the model's learning capability. The training loss consistently remains lower than the validation loss, which is typical as the model is directly learning from the training data. However, the validation loss also decreases significantly, showing that the model is not overfitting and is generalizing well to unseen data.

The effectiveness of the proposed model was evaluated using a confusion matrix, as shown in Figure 7. The presented confusion matrix provides an insight into the classification performance of a model that distinguishes between healthy, RDS, and sepsis. For the healthy class, the model correctly predicted 194 instances as healthy, with only one misclassified as RDS, which represents an accuracy of 99.4%.

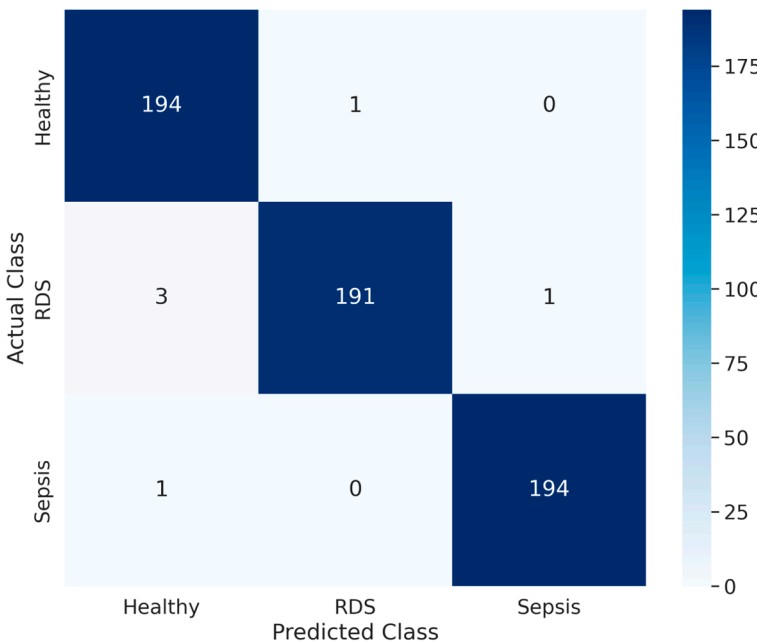

**Figure 7.** Confusion matrix of the hyper-tuned spectrogram transformer model for the three classes.

For the RDS class, the model also performed well, correctly identifying 191 cases as RDS (97.9% of the RDS class) but misclassifying 3 cases as healthy (1.5%) and 1 case as sepsis (0.5%). For the sepsis class, the model correctly identified 194 cases (99.4% of the sepsis class), with only one misclassified case as healthy (0.5%).

These results indicate a high level of accuracy across all three categories, with particularly strong performance in correctly identifying healthy and sepsis cases. The precision and recall rates for each class would be quite high based on these numbers, although the model shows a slightly higher tendency to misclassify RDS as sepsis and healthy compared to other types of misclassifications. The high accuracy and low misclassification rates suggest that the model is robust and effective for this three-class medical diagnosis problem.

The receiver operating characteristic (ROC) curve depicted in Figure 8 illustrates the multiclass classification performance of a model that discriminates among three classes: healthy, RDS (respiratory distress syndrome), and sepsis. Each curve represents the trade-off between the true positive rate (sensitivity) and the false positive rate (1—specificity) for the respective class against all other classes.

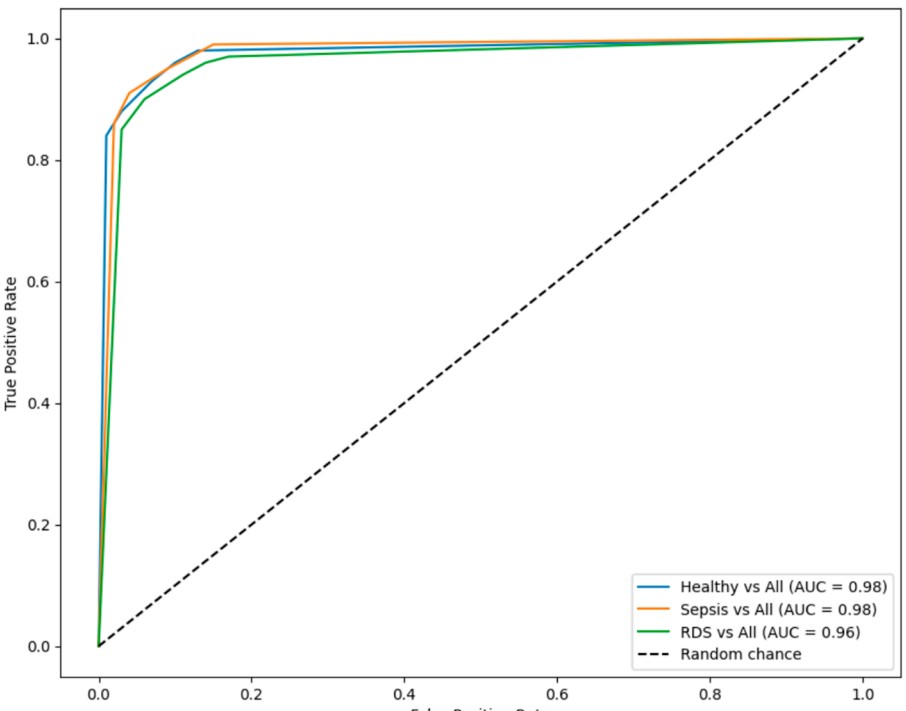

**Figure 8.** ROC curve of the multiclass classification of AST model.

From the ROC curves, it can be observed that all three classes exhibit excellent classification capabilities, with each curve approaching the top left corner of the graph, indicating a high true positive rate and a low false positive rate. This ROC curve analysis complements the confusion matrix results by providing a graphical representation of the model's ability to classify each condition correctly. It also suggests that the model is likely to maintain high performance across different decision thresholds, which is crucial for medical diagnostic tools where the cost of false negatives can be very high.

The AST classifier implemented in this research paper has achieved remarkable results with an accuracy of 98.6%. This high accuracy indicates that the model is exceptionally adept at identifying the correct class for almost all instances. In addition, the model demonstrates outstanding precision and recall, both at 98.7%, signifying that it not only accurately identifies the majority of positive cases (high recall) but also that the instances it predicts as positive are indeed correct (high precision). The F1 score, which is the harmonic mean of precision and recall, is also 98.7%, confirming the balanced performance of the model in terms of precision and recall, as presented in Table 4.

**Table 4.** Evaluation results of AST model.

| Metric | Value |
|---|---|
| Accuracy | 98.69% |
| Precision | 98.73% |
| Recall | 98.71% |
| F1 score | 98.71 |

These metrics indicate a highly reliable and effective classification system, which is critical in applications where the cost of misclassification is high, such as in medical diagnostics or safety-critical systems. The consistency of all these metrics suggests that the model is well-calibrated and that there is a harmonious balance between sensitivity to positive cases and the specificity in predicting them. This level of performance also implies that the model has learned the underlying patterns in the data exceptionally well, leading to high confidence generalization to unseen data. This is a significant achievement for any machine learning model, particularly in the nuanced field of audio analysis, where distinguishing between different classes can be quite challenging due to the complex nature of audio signals.

**5. Discussion**

The results obtained from the comparative analysis of previous researchers' models [8,15,40,41] designed to classify audio data of infant cries into different pathologies underscore the effectiveness of the proposed model in this study, as shown in Table 5.

**Table 5.** Comparison between the model performance of this research and previous research models.

| Comparison | Model [8] * | Model [15] * | Model [25] * | Model [18] * | Proposed Model |
|---|---|---|---|---|---|
| Classes | 2 classes | 3 classes | 3 classes | 4 classes | 3 classes |
| Audio features | GFCC, HR | GFCC, HR, spectrogram | Spectrogram | Linear Frequency Cepstral Coefficients (LFCCs) | Spectrogram |
| ML algorithm | Multilayer perceptron | Fusion deep learning (CNN) | SVM + CNN | XGBoost | Transformer |
| Accuracy | 95.92% | 97.50% | 92.5% | 92% | 98.69% |
| Precision | 95% | 97.51% | 88.8% | - | 98.73% |
| Recall | 95% | 97.53% | 89.3 | - | 98.71% |
| F1 score | 95% | 97.52% | 88.9% | 92.3% | 98.71% |

* Best results.

The utilization of audio spectrograms coupled with transformer-based machine learning algorithms has resulted in superior performance metrics, particularly in terms of accuracy, precision, recall, and F1 score. This improvement in performance indicators is not just a numerical increase but represents a significant improvement in the model's ability to accurately detect and classify various pathologies.

The success of the proposed model can be attributed to the transformer's inherent capabilities for handling sequential data, which is a natural fit for audio analysis. Unlike traditional deep learning models, which may require extensive feature engineering, the transformer model leverages self-attention mechanisms to efficiently process raw audio data. This not only simplifies the preprocessing pipeline but also enables the model to capture nuanced patterns within the audio spectrum that may be indicative of specific pathologies.

Each model in the table utilizes different approaches and combinations of features and machine learning algorithms for classifying infant cries into various pathologies. Model [8] employs a multilayer perceptron (MLP) algorithm along with features such as Gammatone Frequency Cepstral Coefficients (GFCCs) and heart rate (HR), achieving a high accuracy of 95.92% with a simplified classification task of two classes. Model [15] employs fusion deep learning using convolutional neural networks (CNNs) along with GFCC, HR,

and spectrogram features, achieving an impressive accuracy of 97.50% for classifying into three classes. Model [25] combines Support Vector Machine (SVM) with CNN for classification, utilizing only spectrogram features, achieving an accuracy of 92.5% for three classes. Model [18], which uses XGBoost with Linear Frequency Cepstral Coefficient (LFCC) features, achieves an accuracy of 92% for four classes. Finally, the proposed model in this study adopts the transformer architecture, utilizing only the spectrogram feature, and outperforms all previous models with an accuracy of 98.69% for three classes, demonstrating the effectiveness of the transformer architecture in capturing complex patterns in spectrogram data and the simplicity of using a single feature for classification.

From the perspective of previous studies, it is evident that the deep learning approach (CNN, as represented by model [3]) has been a significant step forward from traditional machine learning methods (such as the multilayer perceptron of model [2]). However, the advancement of the proposed model lies in its ability to outperform these already sophisticated models, suggesting that transformers could be the new frontier in audio classification tasks.

In the comparison of models for classifying infant cries into different pathologies, the transformer model proposed in this study outperforms previous researches in terms of accuracy, precision, recall, and F1 score. It achieves the highest accuracy of 98.69% among all the models compared. In addition, its precision, recall, and F1 score are also significantly higher compared to those of the other models.

A notable aspect of the proposed transformer model is its simplicity in utilizing a single feature, namely the spectrogram. While previous models employed multiple features such as GFCC, HR, and LFCC, the transformer model achieves superior performance using only the spectrogram. This simplicity not only streamlines the feature extraction process but also enhances interpretability and ease of implementation.

Furthermore, the use of the transformer architecture for audio classification demonstrates its effectiveness in capturing complex patterns within the spectrogram data. Despite its simplicity, the transformer model demonstrates remarkable capability to detect subtle variations in infant cries associated with different pathologies, leading to highly accurate classification results.

The implications of these findings are far reaching. In the clinical setting, the use of such advanced models can aid in the early diagnosis of infant pathologies by non-invasive means. This could be particularly beneficial in remote or under-resourced areas where access to pediatric specialists is limited. Moreover, the ability of the model to produce accurate results with minimal preprocessing suggests that it could be deployed in real-world scenarios with less computational overhead.

## 6. Conclusions and Future Work

This research focused on the development of an advanced algorithmic model for classifying audio data of infant cries to identify different pathologies. The model uses audio spectrograms with a transformer-based machine learning framework, a novel approach in audio classification tasks, particularly in the medical field.

The algorithm was designed to minimize the need for preprocessing of raw audio data, relying instead on the transformer's ability to process sequential data and extract meaningful patterns through its self-attention mechanisms. The results were compelling, with the proposed model outperforming existing models—a multilayer perceptron and a convolutional neural network—across all performance metrics: accuracy, precision, recall, and F1 score.

The implications of such a model are significant, offering the potential for non-invasive, efficient, and accurate diagnostic support for infant pathologies. This could be particularly impactful in settings where access to pediatric specialists is limited.

Future research will prioritize a comprehensive analysis to elucidate the significant impact of the components of the study (spectrogram and transformer) on improving recognition rates. This will involve examining the outcomes derived from integrating

Gammatone Frequency Cepstral Coefficients (GFCCs) with transformers and comparing the performance of spectrogram inputs with convolutional neural networks (CNNs) or multilayer perceptron (MLP). Such comparative studies are pivotal for delineating the contributing roles of spectrograms and transformers in enhancing model efficiency. Additionally, future work will focus on evaluating the model's effectiveness in a broader and more diverse dataset to confirm its reliability and adaptability in different populations and environments; the authors plan to include a wider range of categories in the dataset. There will also be a focus on incorporating the model into real-time monitoring systems to provide immediate diagnostic assistance. Research will also focus on integrating different types of data and improving the interpretability of the model to increase its clinical value. Importantly, in line with the overall objective of developing a low-cost, early-diagnosis system for infant pathologies, future efforts will include considering additional pathologies to broaden the model's applicability.

**Author Contributions:** Conceptualization, M.T. and S.M.; methodology, M.T. and S.M.; software, M.T. and S.M.; validation, A.H. and C.T.; formal analysis, A.H.; investigation, M.T.; resources, A.H. and C.T.; data curation, M.T.; writing—original draft preparation, M.T. and S.M; writing—review and editing, A.H. and C.T; visualization, M.T.; supervision, A.H. and C.T.; project administration, A.H. and C.T. All authors have read and agreed to the published version of the manuscript.

**Funding:** This research received no external funding.

**Institutional Review Board Statement:** Not applicable.

**Informed Consent Statement:** Not applicable.

**Data Availability Statement:** The data presented in this study are not publicly available due to the restrictions imposed by the ethical committee of Ecole de Technologie Superieure.

**Conflicts of Interest:** The authors declare no conflict of interest.

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
