# Peer review of "Transformer-Based Approach to Pathology Diagnosis Using Audio Spectrogram"

_information, doi:10.3390/info15050253_

Round 1

Reviewer 1 Report

Comments and Suggestions for Authors

Recognizing pathological signs based on a baby's crying sound is a very important application. In this paper,  a method to convert the baby's crying sound into a spectrogram and apply a transformer to recognize three patterns (Healthy, RDS, and Sepsis) was proposed. Given the importance of the application and the application of state-of-the-art AI techniques (transformer), I believe that this work is of interest to many researchers and deserves to be published. However, it would be desirable that the following points should be reviewed and revised by the author before publication.

Page 2, line 79. It's worth mentioning what GFCC and HR stand for first. An explanation of these two acronyms was found on page 15, line 562, but they should be explained the acronym is firstly mentioned. 

Page 4, line 161. The authors mentioned that complex processing could be avoided from feature extraction by applying the proposed approach. However, the Fourier transforms required for computing GFCC and MFCC can be computed with relatively few computations using high-speed algorithms. 

Page 4. Section 3.1

How many children participated in the experiment? How many children were healthy and how many were sick?

Page 5, line 207~209

In the paper, it was mentioned that 2000 samples were used per category, but in the next sentence it was explained that 1300 samples were used for each class and 3900 segments. Clarification is needed on exactly how many samples and segments were used for each class (healthy, Sepsis,. RDS).

Page 6, lines 249~255.

Detailed conditions (e.g. FFT length, window type, window length, and window shift) used in the spectrogram calculation should be presented.  

Page 7, lines 290~300

One of the key ideas in this research seems to be the application of ASTs. While the usefulness of AST was validated by the experimental results in this study, it is necessary for the authors to first present the rationale for applying AST.

Page 9, Section 3.5

What loss function was used to train the neural network? 

It was concerned that the best performance at only 6 epochs might be due to the lack of more diverse patterns in the training data. 

Page 15, Table 5.

Were the results for each method obtained using the same training/testing dataset? 

The main focus of the paper seems to be application of spectrograms to baby crying sound and the usage of transformers for recognition. The authors need to analyze which of these two (spectrograms and transformer) played a more important role in improving the recognition rate. As an example, presenting the results of the GFCC+transformer combination and the spectrogram+CNN (or MLP) combination in Table 5 would give a better understanding of the relative importance of spectrograms and transformers.  

Author Response

The authors of the manuscript would like to express their gratitude to the reviewer for insightful feedback, which has helped to improve the manuscript. We hope that our detailed responses to the reviewers’ comments, outlined below, will further clarify the improvements made to the manuscript and address any concerns raised. All comments are also reflected in the updated manuscript.

Comment 1: Page 2, line 79. It's worth mentioning what GFCC and HR stand for first. An explanation of these two acronyms was found on page 15, line 562, but they should be explained the acronym is firstly mentioned.

Response: The explanation of GFCC and HR has been updated in our manuscript where these two acronyms were first mentioned on page 2, line 79. The manuscript explanation now becomes: “The study utilizes only the spectrogram without extracting complicated features such as Gammatone Frequency Cepstral Coefficients (GFCC) and Heart Rate (HR) from the voice”.

Comment 2: Page 4, line 161. The authors mentioned that complex processing could be avoided from feature extraction by applying the proposed approach. However, the Fourier transforms required for computing GFCC and MFCC can be computed with relatively few computations using high-speed algorithms.

Response: We agree with the reviewer about the relatively few computations using high-speed algorithms to construct GFCC. In our manuscript, we aim to describe this using a transformer; our proposed method depends only on a single modality (Spectogram) to achieve the high result as presented in our manuscript, rather than using multiple features i.e. GFCC, HR, and spectrogram. The manuscript on Page 4, line 161 has been updated to reflect this concept: The proposed approach relies solely on the use of a spectrogram from the infant cry audio dataset instead of using multiple features such as GFCC and HR to achieve high performance results.

Comment 3: Page 4. Section 3.1 - How many children participated in the experiment? How many children were healthy and how many were sick?

Response: The manuscript has been updated to include the number of infant participants in each of the three categories. There were a total of 53 recordings from 17 newborns for Sepsis group, 102 recordings from 33 newborns for the RDS pathology group, and 181 recordings from 83 newborns for the Healthy group.

Comment 4: Page 5, line 207~209 - In the paper, it was mentioned that 2000 samples were used per category, but in the next sentence it was explained that 1300 samples were used for each class and 3900 segments. Clarification is needed on exactly how many samples and segments were used for each class (healthy, Sepsis, RDS).

Response: The mention of 2000 samples in the manuscript was a mistake. The correct number of samples for each class is 1300. This correction is consistent with the confusion matrix in Figure 7, which shows that each class from the test dataset (15%) comprises exactly 195 samples, aligning with the calculation 1300*0.15 = 195. The manuscript was updated to accurately reflect the correct number of samples per class.
The dataset was then evenly sampled to ensure an equal representation of samples from each category. This approach yielded a balanced and uniform dataset containing 3900 segmented records with 1300 samples per each class, as shown in Figure 2.

Comment 5: Page 6, lines 249~255 - Detailed conditions (e.g. FFT length, window type, window length, and window shift) used in the spectrogram calculation should be presented.

Response: The transformation used to construct a spectrogram from the infant cray audio employing an FFT (Fast Fourier Transform) length of 2048, a Hann window type for windowing, a window length equal to the FFT length (2048), and a hop length (window shift) of 512. The manuscript has been updated to incorporate a detailed discussion of these parameters, along with an analysis of how these parameters might affect the performance of the model.

Comment 6: Page 7, lines 290~300 - One of the key ideas in this research seems to be the application of ASTs. While the usefulness of AST was validated by the experimental results in this study, it is necessary for the authors to first present the rationale for applying AST.

Response: The manuscript has been revised to include the reasoning for selecting the Audio Spectrogram Transformer (AST) for application to health audio data, and to reference three recent research studies that have chosen to use the spectrogram AST on the audio data for high results.

The decision to employ Audio Spectrogram Transformer (AST) models was based on their proven effectiveness at processing complex audio patterns. This ability is particularly beneficial when dealing with the intricate and subtle variations found in health-related audio data. Recent research leveraging AST models with spectrograms has shown promising results, as evidenced in recent studies [40,41,42]. This success has led us to apply the AST model to health-related audio data, which is, to the best of our knowledge, its first application in the health field. The authors aim to leverage its superior pattern recognition capability for a multiclass classification challenge by fine-tuning the AST model with specific health-related audio data. This strategy not only exploits into the unique strengths of AST for processing audio data, but also explores its potential to significantly improve the analysis and categorization of audio datasets in the health domain.

Comment 7: Page 9, Section 3.5 - What loss function was used to train the neural network? It was concerned that the best performance at only 6 epochs might be due to the lack of more diverse patterns in the training data.

Response: Categorical (Multiclass) Cross-Entropy was used in our experiment. The manuscript has been updated to reflect this choice. We have also added it to Table 4.
In this context, Categorical (Multiclass) Cross-Entropy was chosen as the utilized loss function due to its effectiveness in handling multi-class classification problems, which is inherent in our audio dataset. This loss function is particularly suitable for scenarios where each input is intended to be classified into one out of many possible categories. It works by comparing the predicted probability distribution across all classes with the actual distribution, where the true class has a probability of 1 and all others probability 0. This approach incentivizes the model to not only predict the correct class, but to also be confident in its prediction, thereby enhancing the overall accuracy of the model.

In our experiment, the peak performance was achieved at epoch 6, which is consistent with the results reported in the original AST paper. The authors of that paper noted that they achieved high results with a relatively low epoch count of 5. In addition, as can be seen in Figure 6, the loss function value of both the training and validation datasets decreased together until epoch 6, where the validation loss started to increase, indicating that the model converged when epoch 6 was reached.

Comment 8: Page 15, Table 5. - Were the results for each method obtained using the same training/testing dataset?

Response: The results presented in the first two studies [8, 15], along with those of the current study, were all derived using the same dataset and the same training/testing protocol. As for the other two studies [43, 44], they utilized different datasets. We have included them in our comparative analysis to introduce more variety to the comparison.

Comment 9: The main focus of the paper seems to be application of spectrograms to baby crying sound and the usage of transformers for recognition. The authors need to analyze which of these two (spectrograms and transformer) played a more important role in improving the recognition rate. As an example, presenting the results of the GFCC+transformer combination and the spectrogram+CNN (or MLP) combination in Table 5 would give a better understanding of the relative importance of spectrograms and transformers.

Response: We agree with the reviewer that further investigation is needed to understand the important role of each part of the current study (spectrogram, transformer) in enhancing the recognition rate. This will be investigated in our future works. We have included this aspect in our future work, which is discussed in Section 6: Conclusions and Future Work.

Future research will prioritize a comprehensive analysis to elucidate the significant impact of the components of the study (spectrogram and transformer) on improving recognition rates. This will involve examining the outcomes derived from integrating Gammatone Frequency Cepstral Coefficients (GFCC) with transformers, and comparing the performance of spectrogram inputs with Convolutional Neural Networks (CNN) or Multi-Layer Perceptrons (MLP). Such comparative studies are pivotal for delineating the contributing roles of spectrograms and transformers in enhancing model efficiency.

Reviewer 2 Report

Comments and Suggestions for Authors

This is a well-written paper presenting interesting results on using the latest transformer model to classify infant crying audios. It includes sufficient details to justify the superior performance of the new method over other published ones. I have the following comments:

1. It is not quite exactly clear why the transformer model outperformed the other ML models, even though some discussions are provided (sequential processing, self-attention, etc.). Further investigation would allow us to have a better understanding. 

2. In this study, the crying sounds are classified into three categories. If more categories are involved, it is unclear if the same level of performance can be maintained with the transformer-based model. 

Author Response

The authors of the manuscript would like to express their gratitude to the reviewer for insightful feedback, which has helped to improve the manuscript. We hope that our detailed responses to the reviewers’ comments, outlined below, will further clarify the improvements made to the manuscript and address any concerns raised. All comments are also reflected in the updated manuscript.

Comment 1: It is not quite exactly clear why the transformer model outperformed the other ML models, even though some discussions are provided (sequential processing, self-attention, etc.). Further investigation would allow us to have a better understanding.

Response:
We agree with the reviewer that further research is needed to better understand whether if the transformer-based model outperforms traditional ML and Deep learning models i.e. CNN in a more complex and diverse environment and dataset. Meanwhile, the high performance revealed of the AST model compared to previous research as shown in Table 5 in the Discussion section could be attributed to the fact that it has self-attention mechanisms, allowing it to focus on relevant parts of the input without losing information, unlike CNNs which may lose fine details due to spatial reduction through convolution and pooling layers.
The manuscript has been revised to include this idea:

The AST model processes spectrograms through self-attention mechanisms, allowing it to focus on relevant parts of the input without losing information, unlike CNNs which can lose fine details due to spatial reduction through convolution and pooling layers. Transformers compute relationships between all parts of the input simultaneously, efficiently capturing both global and local dependencies. This ability to maintain a comprehensive understanding of the entire input at each layer enables more nuanced and detailed modeling of audio data, leading to its superior performance.

Comment 2: In this study, the crying sounds are classified into three categories. If more categories are involved, it is unclear if the same level of performance can be maintained with the transformer-based model.

Response:

In order to provide more robust support for the results presented in this manuscript, the authors intend to expand the dataset to include additional classes to ensure the scalability of the model and maintain its performance across a broader range of categories. This intention is included in the Conclusions and Future Work section.

Additionally, future work will focus on evaluating the model's effectiveness in a broader and more diverse dataset to confirm its reliability and adaptability in different populations and environments, the authors plan to include a wider range of categories in the dataset.

Reviewer 3 Report

Comments and Suggestions for Authors

Although the work reported in the manuscript is technically correct, it is essentially an application of a machine learning methodology, without novelty in the principles, in the processing, or in the interpretation of the results. Therefore, the impact of the work is more on the engineering side than on the research side. In conclusion, I don't think the manuscript suitable for publication in an archival journal.

Author Response

The authors of the manuscript would like to express their gratitude to the reviewer for insightful feedback. We hope that our responses to the reviewer's concern, outlined below, will further clarify the importance of our work and address any concerns raised.

Comment 1: Although the work reported in the manuscript is technically correct, it is essentially an application of a machine learning methodology, without novelty in the principles, in the processing, or in the interpretation of the results. Therefore, the impact of the work is more on the engineering side than on the research side. In conclusion, I don't think the manuscript suitable for publication in an archival journal.

Response:

Thank you for taking the time to review our manuscript. We appreciate your feedback and the opportunity to address your concerns regarding the novelty and impact of our work.

While we understand your perspective, we respectfully disagree with the assessment that our manuscript lacks novelty in the principles, processing, or interpretation of the results. We believe that our work makes several significant contributions to the field of machine learning for audio analysis, particularly in the context of infant cry classification.

Firstly, our use of a transformer-based algorithm for audio spectrogram analysis represents a novel approach in the domain of infant cry analysis. Transformers have traditionally been applied to natural language processing tasks and have shown remarkable success. Our adaptation of this architecture to the analysis of audio signals, specifically in distinguishing between different pathologies in infant cries, presents a novel and innovative application of transformer technology.

Secondly, unlike previous models that often require different feature engineering and preprocessing, our model streamlines the process by leveraging the transformer's self-attention mechanisms to extract meaningful patterns directly from raw audio spectrograms. This approach not only simplifies the computational pipeline but also enhances the interpretability of the model's predictions, which is crucial in medical diagnostic applications.

Furthermore, our comparative analysis with existing models [8, 15, 43, 44] demonstrates that our proposed model outperforms these models in terms of accuracy, precision, recall, and F1-score. This improvement in performance metrics signifies a significant advancement in the field and underscores the impact of our work.

In conclusion, we believe that our manuscript represents a substantial research contribution, showcasing the innovative application of transformer-based machine learning in audio analysis, particularly for the classification of infant crying signals. We respectfully request that you reconsider your evaluation and recognize the novel and impactful nature of our work.

Round 2

Reviewer 1 Report

Comments and Suggestions for Authors

The authors have carefully checked all the points issued in the first round review and revised the paper accordingly. 

Therefore, the reviewer recommends that this article be accepted for publication.

Reviewer 2 Report

Comments and Suggestions for Authors

ready to be accepted.